# Global Sensitivity of Tropospheric Ozone to Precursor Emissions in Clean and Present-Day Atmospheres: Insights from AerChemMIP Simulations

Wei Wang[1] and Chloe Yuchao Gao[2,3,*]

[1] Nanjing-Helsinki Institute in Atmospheric and Earth System Sciences, Nanjing University, Nanjing, 210023, China

[2] Department of Atmospheric and Oceanic Sciences & Shanghai Key Laboratory of Ocean-Land-Atmosphere Boundary Dynamics and Climate Change, Fudan University, Shanghai, 200438, China

[3] Institute of Eco-Chongming (IEC), Shanghai, 202151, China

[*] Corresponding author: Chloe Yuchao Gao (gyc@fudan.edu.cn)

**Abstract**

14          Ozone ($O_3$) is a Short-lived Climate Forcer (SLCF) that contributes to radiative

forcing and indirectly affects the atmospheric lifetime of methane, a major
greenhouse gas. This study investigates the sensitivity of global $O_3$ to precursor
gases in a clean atmosphere, where hydroxyl (OH) radical characteristics   are more
spatially uniform than in present-day conditions, using data from the *PiClim*
experiments of the Aerosols and Chemistry Model Intercomparison Project
(AerChemMIP) within the CMIP6 framework. We also evaluate the $O_3$ simulation
capabilities of four Earth system models (CESM2-WACCM, GFDL-ESM4, GISS-
E2-1-G, and UKESM1-0-LL). Our analysis reveals that the CESM and GFDL
models effectively capture seasonal $O_3$ cycles and consistently simulate vertical $O_3$
distribution. While all models successfully simulate $O_3$ responses to anthropogenic
precursor emissions, CESM and GFDL show limited sensitivity to enhanced natural
$NO_x$ emissions (e.g., from lightning) compared to GISS and UKESM. The
sensitivities of $O_3$ to its natural precursors ($NO_x$ and VOCs) in GISS and UKESM
models are substantially lower than their responses to anthropogenic emissions,
particularly for lightning $NO_x$ sources. These findings refine our understanding of
$O_3$ sensitivity to natural precursors in clean atmospheres and provide insights for
improving $O_3$ predictions in Earth system models.

## 1 Introduction

Tropospheric ozone ($O_3$) is a key air pollutant and atmospheric oxidant, exerting extensive influence on air quality and human health (Coffman et al., 2024; Lim et al., 2019; Malley et al., 2017; Nuvolone et al., 2018), climate systems, and biogeochemical processes (Hu et al., 2023; Fowler et al., 2009). As a Short-lived Climate Forcer (SLCF), tropospheric $O_3$ exerts a radiative forcing of 0.35–0.5 W m$^{-2}$ and influences atmospheric processes such as evaporation, cloud formation, and general circulation (Khomsi et al., 2022; Möller and Mauersberger, 1992; Rogelj et al., 2014; Stevenson et al., 2013). Furthermore, $O_3$ plays a crucial role in regulating the terrestrial carbon sink and enhancing the formation of the hydroxyl (OH) radical (Naik et al., 2013b), which, in turn, affect the lifetime of methane (and halocarbons), the second most prominent anthropogenic greenhouse gas after carbon dioxide (Kumaş et al., 2023). $O_3$ also contributes to an increased atmospheric oxidation capacity, influencing the formation of secondary aerosols, such as organic aerosol, sulfate, and nitrate, which have significant implications for radiative forcing (Karset et al., 2018).

While stratospheric $O_3$ entrainment contributes to tropospheric $O_3$ levels, the primary source of tropospheric $O_3$ is photochemical production. This secondary pollutant is formed through photochemical oxidation reactions involving oxides of nitrogen ($NO + NO_2 = NO_x$) and volatile organic compounds (VOCs) in the presence of OH and hydroperoxyl ($HO_2$) radicals (Monks et al., 2015). The relationship between $O_3$ and its precursors is nonlinear, making it challenging to mitigate $O_3$ pollution through simple precursor reduction strategies. Regional-scale sensitivity to $O_3$ precursors has been extensively investigated, such as emphasizing the diagnostic utility of ratios including $O_3/NO_x$ (Jin et al., 2023; Sillman and He, 2002) and $VOC/NO_x$ (Li et al., 2024) for assessing $O_3$-$NO_x$-VOC sensitivity, and nations such as the United Kingdom and the United States have demonstrated significant success in controlling regional ozone levels by implementing measures to reduce $NO_x$ emissions (Hakim et al., 2019). However, the global-scale sensitivity of $O_3$ to its precursors has received limited attention, despite evidence suggesting that global $O_3$ forcing may have a more substantial impact on climate forcing than localized $O_3$ enhancements. Consequently, improving our understanding of $O_3$ formation

mechanisms on a global scale is essential for effective air quality management and climate change mitigation strategies (Yu et al., 2021).

Recent studies utilizing Coupled Model Intercomparison Project Phase 6 (CMIP6; Eyring et al., 2016) datasets have offered insights into the spatio-temporal evolution of the global tropospheric $O_3$ budget from 1850 to 2100 (Griffiths et al., 2021; Turnock et al., 2019) and have quantified the global stratosphere-troposphere $O_3$ exchange process (Li et al., 2024; Griffiths et al., 2021). However, challenges persist in quantifying the sensitivity of global $O_3$ to its precursors when assessing the increasing global $O_3$ forcing attributed to these precursors. These challenges arise from regional variability in meteorological conditions (Carrillo-Torres et al., 2017), differences in $NO_x$ and VOC volume mixing ratios (Jin et al., 2023; Sillman and He, 2002), and the distinct characteristics of OH and $HO_2$ influenced by varying degrees of urbanization (Karl et al., 2023; Vermeuel et al., 2019). Furthermore, while the observed upward trends in $O_3$ levels are primarily attributed to increased precursor emissions, limited research has investigated whether contemporary atmospheric conditions—shaped by climate warming and enhanced oxidation capacities—may be creating a more favorable environment for $O_3$ formation.

To address these gaps, this study investigates the sensitivity of global-scale $O_3$ to its precursors under a pre-industrial background atmosphere, with approximate uniform $HO_x$ conditions in major continental areas. We also examine the feedback mechanisms of different model responses to precursors from both anthropogenic and natural sources, using *PiClim* experiment data from the Aerosols and Chemistry Model Intercomparison Project (AerChemMIP) simulations (Collins et al., 2017) within CMIP6. Additionally, this research evaluates the ozone formation potential in the pre-industrial era based on contemporary (2014) emissions of $O_3$ precursors, with the aim of elucidating whether shifts in the background atmosphere have rendered it chemically more conducive to $O_3$ generation. Our analysis employs four models with interactive stratospheric and tropospheric chemistry, which have been extensively utilized in $O_3$-related research (Brown et al., 2022; Griffiths et al., 2021; Tilmes et al., 2022; Zeng et al., 2022). This approach allows us to assess the global-scale sensitivity of $O_3$ to its precursors, evaluate the consistency and discrepancies among different models in representing $O_3$-precursor relationships, and provide insights into

the potential impacts of changing emissions on future global $O_3$ levels and associated
climate forcing, contributing to more accurate projections of future climate change.
**2 Models and methods**
**2.1 Model descriptions**
We use monthly-mean simulation data from four Earth system models in this
study. The four chosen models possess the benefit of extensive applicability and a
comprehensive *PiClim* computational framework. Table 1 summarizes key model
features, including model resolution, vertical stratification, complexity of gas-phase
chemistry, and relevant references. All models include interactive coupling of
tropospheric and stratospheric chemistry with $O_3$ dynamics integrated into the
radiation scheme, simulating the interaction between $O_3$ concentration and
temperature. The response of simulated reactive gas emissions to chemical
complexity is important. For example, changes in Biogenic Volatile Organic
Compounds (BVOCs) can impact $O_3$, methane lifetime, and potentially the oxidation
of other aerosol precursors in models with interactive tropospheric chemistry via OH
changes.
**Table 1.** Information on model resolution, vertical levels, property of gas-phase chemistry and references.

| Model | Resolution (lat × lon) | Number of gridpoints | Vertical levels | Aerosol model | Simulation reference |
|---|---|---|---|---|---|
| CESM2-WACCM | 192 × 288 | 55296 | 70 levels; top level $6 \times 10^{-6}$ hPa | MAM4 | (Gettelman et al., 2019) |
| GFDL-ESM4 | 180 × 288 | 51840 | 49 levels; top level 0.01 hPa | MATRIX | (Dunne et al., 2020; Horowitz et al., 2020) |
| GISS-E2-1-G | 90 × 144 | 12960 | 40 levels; top level 0.1 hPa | OMA | (Miller et al., 2014; Kelley et al., 2020) |
| UKESM1-0-LL | 144 × 192 | 27648 | 85 levels; top level 1 hPa | GLOMAP | (Mulcahy et al., 2018; Sellar et al., 2019) |


CESM2-WACCM (hereafter "CESM") is a fully coupled Earth system model
that integrates the Community Earth System Model version 2 (Emmons et al., 2020)
with the Whole Atmosphere Community Climate Model version 6 (WACCM6). The
atmospheric component operates at a horizontal resolution of $0.9375°$ latitude by $1.25°$
longitude, with 70 hybrid sigma-pressure vertical layers extending from the surface
to $6 \times 10^{-6}$ hPa. Its interactive chemistry and aerosol modules include the troposphere,
stratosphere, and lower thermosphere, with a comprehensive treatment of 231 species,
photolysis reactions, 403 gas-phase reactions, 13 tropospheric heterogeneous
reactions, and 17 stratospheric heterogeneous reactions (Emmons et al., 2020). The
model utilizes the four-mode Modal Aerosol Model (MAM4) (Emmons et al., 2020)
and features its secondary organic aerosol (SOA) framework based on the Volatility
Basis Set (VBS, Donahue et al., 2013) approach. The photolytic calculations use both
inline chemical modules and a lookup table approach, which does not consider
changes in aerosols.
The Atmospheric Model version 4.1 (AM4.1, Horowitz et al. (2020)) within the
GFDL Earth system model (Dunne et al., 2020) incorporates an interactive chemistry
scheme that spans both the troposphere and stratosphere (GFDL-ESM4; hereafter
"GFDL"). The atmospheric component operates at a horizontal resolution of $1°$
latitude by $1.25°$ longitude, with 49 hybrid sigma-pressure vertical layers extending
from the surface to 0.01 hPa. This scheme includes 56 prognostic tracers, 36
diagnostic species, 43 photolysis reactions, 190 gas-phase kinetic reactions, and 15
heterogeneous reactions. Stratospheric chemistry accounts for key $O_3$ depletion
cycles ($O_x$, $HO_x$, $NO_x$, $ClO_x$, and $BrO_x$) and heterogeneous reactions on stratospheric
aerosols (Austin et al., 2013). Photolysis rates are calculated dynamically with the
FAST-JX version 7.1 code, which considers the radiative impacts of modeled
aerosols and clouds. The chemical mechanism is further elaborated in Horowitz et al.
(2020), and the gas-phase and heterogeneous chemistry are similar to those employed
by Schnell et al. (2018). Non-interactive natural emissions of $O_3$ precursors are
prescribed as outlined in Naik et al. (2013a).
The GISS model, developed by the NASA Goddard Institute for Space Studies,
integrates the chemistry-climate model version E2.1 with the GISS Ocean v1 (G01)
model (GISS-E2-1-G; hereafter "GISS"). The specific configurations of this model

utilized for the CMIP6 are detailed in Kelley et al. (2020). In this study, we focus on the model subset that includes online interactive chemistry. The atmospheric component operates at a horizontal resolution of 2° latitude by 2.5° longitude, with 40 hybrid sigma-pressure vertical layers extending from the surface to 0.1 hPa. The interactive chemistry module employs the GISS Physical Understanding of Composition-Climate Interactions and Impacts (G-PUCCINI) mechanism for gas-phase chemistry (Kelley et al., 2020; Shindell et al., 2013). For aerosols, the model utilizes either the One-Moment Aerosol (OMA) or the Multiconfiguration Aerosol Tracker of Mixing state (MATRIX) model (Bauer et al., 2020). The gas-phase chemistry involves 146 reactions, including 28 photodissociation reactions, affecting 47 species across the troposphere and stratosphere, along with an additional five heterogeneous reactions. The model transports 26 aerosol particle tracers and 34 gas-phase tracers (OMA).

UKESM represents the United Kingdom's Earth system model (Sellar et al., 2019). It builds upon the Global Coupled 3.1 (GC3.1) configuration of HadGEM3 (Williams et al., 2018), incorporating additional Earth system components, such as ocean biogeochemistry, the terrestrial carbon-nitrogen cycle, and atmospheric chemistry (UKESM1-0-LL; hereafter "UKESM"). Walters et al. (2019) provided descriptions of the atmospheric and land components. The atmospheric component operates at a horizontal resolution of 1.25° latitude by 1.875° longitude, with 85 vertical layers extending from the surface to 85 km. The chemistry module in the UKESM model is a unified stratosphere-troposphere scheme (Archibald et al., 2020) including 84 tracers, 199 bimolecular reactions, 25 unimolecular and termolecular reactions, 59 photolytic reactions, 5 heterogeneous reactions, and 3 aqueous-phase reactions for the sulfur cycle from the United Kingdom Chemistry and Aerosols (UKCA) model. The aerosol module is based on the two-moment scheme from UKCA, known as GLOMAP mode, and is integrated into the Global Atmosphere 7.0/7.1 configuration of HadGEM3 (Walters et al., 2019). The UKESM uses interactive Fast-JX photolysis scheme, which is applied to derive photolysis rates between 177 and 850 nm, as described in Telford et al. (2013). In the lower mesosphere, photolysis rates are calculated using lookup tables (Lary and Pyle, 1991).

Models differ in their representation of $O_3$ source and sink processes, as well as in the definitions of the associated budget terms, which contributes to variability in

model outcomes (Stevenson et al., 2006; Young et al., 2018). For example, in the
GISS model, the tropospheric chemistry component simulates the $NO_x$-$HO_x$-$O_x$-CO-
$CH_4$ system and the oxidation pathways for non-methane volatile organic compounds
(NMVOCs). Central to these discrepancies are the treatments of non-methane volatile
organic compound NMVOCs chemistry, which impacts both chemical production
and destruction rates, along with surface removal mechanisms and stratospheric
influences. Furthermore, the choice of tropopause definition can significantly alter
the diagnosed $O_3$ burden, as well as the flux from the stratosphere.
All four of the interactive tropospheric chemistry models contain
parameterizations of the nitrogen oxide ($NO_x$) emissions from lightning based on the
height of the convective cloud top (Price et al., 1997; Price and Rind, 1992; Price,
2013), and the tropopause height for each model based on the WMO definition. Each
model has a different way of implementing emissions and how much they are profiled.
For instance, online calculations of lightning $NO_x$ emissions during deep convection
in the GISS model are based on the method described by (Kelley et al., 2020).
Lightning $NO_x$ continues to be a major source of uncertainty in both model
comparisons and the temporal development of tropospheric $O_3$ because it has a
disproportionately significant influence on tropospheric-$O_3$ concentration relative to
surface emissions (Murray et al., 2013).
BVOC emissions are modeled as a function of vegetation type and cover, as well
as temperature and photosynthetic rates (gross primary productivity) (Unger, 2014;
Sporre et al., 2019; Pacifico et al., 2011; Guenther et al., 1995). While models vary
in the speciation of emitted VOCs, they commonly include isoprene and
monoterpenes, each with its own distinct emission parameterization. Despite the
common reliance on photosynthetically active radiation for the parameterization of
BVOC emissions across the four models, there exist notable distinctions. For instance,
the GFDL model exclusively considers the leaf area index, neglecting the impact of
temperature on BVOC emissions, and the CESM, GISS, and UKESM models omit
the influence of vegetation type from their calculations.
**2.2 Simulation data and experimental design**
The primary objective of AerChemMIP is to quantitatively ascertain the
influence of aerosols and reactive trace gases on the climate system, as well as the
bidirectional feedback mechanisms involved (Collins et al., 2017). Table 2 presents
a synopsis of the experimental configurations employed in this study. The control
experiment, denoted as *PiClim-control*, is designed to stabilize both atmospheric
composition and climatic conditions at a state reminiscent of the pre-industrial era,
where the natural fractions of stratospheric ozone forcing species such as halocarbons
was extremely low, specifically 1850. The *PiClim-2x* experiment involves doubling
of individual natural emission fluxes relative to the 1850 control, while the *PiClim-x*
experiments calibrate these fluxes to align with the emission levels prevalent in 2014
(Collins et al., 2017). *PiClim-2xNO$_x$* represents to doubling of the nitric oxide
emissions from natural sources due to lightning activity. *PiClim-2xVOC* represents to
doubling of the volatile organic compound emissions from natural sources, including
isoprene and monoterpenes. *PiClim-HC* represents the pre-industrial climatological
control with 2014 halocarbons emissions both from anthropogenic (CFCs, HCFCs
and compounds containing bromine) and natural sources. *PiClim-CH$_4$* represents the
pre-industrial climatological control with 2014 methane emissions both from
anthropogenic and natural sources. *PiClim-NO$_x$* represents the pre-industrial
climatological control with 2014 nitrogen oxide emissions both from anthropogenic
and natural sources. *PiClim-VOC* represents the pre-industrial climatological control
with 2014 VOC emissions both from anthropogenic and natural sources. *PiClim-*
*NTCF* represents the pre-industrial climatological control with 2014 near-term
climate forcers emissions, including aerosols and chemically reactive gases such as
tropospheric ozone and methane. *PiClim-N$_2$O* represents the pre-industrial
climatological control with 2014 nitrous oxide emissions both from anthropogenic
and natural sources. *PiClim-aer* represents the pre-industrial climatological control
with 2014 aerosol concentrations. *PiClim- O$_3$* represents the pre-industrial
climatological control with 2014 ozone concentrations. *PiClim-BC* represents the pre-
industrial climatological control with 2014 black carbon concentrations.

**Table 2.** The available experiments of selected models in this study. "X" represents the experiment is available

| PiClim- / Model | 2xNO$_x$ | 2xVOC | HC | CH$_4$ | NO$_x$ | VOC | NTCF | N$_2$O | O$_3$ | aer | control | BC |
|---|---|---|---|---|---|---|---|---|---|---|---|---|
| CESM2-WACCM | X | X | X | X | X | X | X | X | | | | |
| GFDL-ESM4 | X | X | X | | X | X | | | X | X | X | X |
| GISS-E2-1-G | X | X | X | X | X | X | X | X | X | X | X | X |
| UKESM1-0-LL | X | X | X | X | X | X | X | X | X | X | X | X |


We analyzed models that had archived sufficient data in the Earth System Grid
Federation (ESGF) system to permit accurate characterization of tropospheric $O_3$. In
practice this meant we used archived $O_3$ data from the AERmon characterization of
the tropospheric $O_3$ (variable name: "o3") on native model grids. Other variables used
include chemical production (variable name: "o3prod"), chemical destruction
(variable name: "o3loss"), nitrogen monoxide (variable name: "no"), nitrogen
dioxide (variable name: "no2"), isoprene (variable name: "isop"), organic dry aerosol
(variable name: "emioa"), and secondary organic aerosol (variable name: "mmrsoa").
All data used in this paper are available on the Earth System Grid Federation website
and can be downloaded from https://esgf-index1.ceda.ac.uk/search/cmip6-ceda/ (last
access: 4 July 2024, ESGF-CEDA, 2020).

A new set of historical anthropogenic emissions has been developed with the
Community Emissions Data System (CEDS, Hoesly et al., 2018). CEDS uses updated
emission factors to provide monthly emissions of the major aerosol and trace gas
species over the period 1750 to 2014 for use in CMIP6, and biomass burning
emissions are based on a different inventory developed separate from CEDS (Van
Marle et al., 2017). The primary analysis examines emissions of $NO_x$ and VOCs from
anthropogenic (Hoesly et al., 2018) and biomass burning sources (van Marle et al.,
2017) that were provided as a common emission inventory to be used by all models
(including the four in this study) in CMIP6 simulations. In the CESM and GFDL
models, biogenic emissions, including isoprene and monoterpenes, are calculated
interactively using MEGAN version 2.1 (Guenther et al., 2012) and are further
utilized for SOA formation. While in the GISS model, biogenic emissions of isoprene
are computed online and are sensitive to temperature (Shindell et al., 2006), whereas
alkenes, paraffins, and terpenes are prescribed. And in the UKESM model, emissions
of isoprene and monoterpenes are interactively calculated using the iBVOC emission
model (Pacifico et al., 2011).
**3 Results and Discussions**
**3.1 Spatial, seasonal, and vertical distribution of tropospheric $O_3$**

We first investigate the seasonal and vertical variations of ozone volume mixing
ratio in the pre-industrial atmospheres simulated by four selected models. The
analysis of tropospheric $O_3$ data derived from the *PiClim* experiment outcomes of

CMIP6 models reveals distinct seasonal cycles and inter-model variations (Fig. 1). The GISS model demonstrates the highest simulated tropospheric column $O_3$ volume mixing ratio at 50.29 ppbv in the 29[th] and 30[th] year of simulation, followed by the UKESM (44.50 ppbv), CESM (38.02 ppbv), and GFDL (31.03 ppbv), where the height of the tropopause is based on the definition of WMO. These are consistent with previous findings from historical experiments (Griffiths et al., 2021).

Furthermore, our analysis indicates that the disparity in $O_3$ volume mixing ratio during the *PiClim* experiment primarily occurs in polar regions. This may be attributed to the GISS model's ability to replicate a more robust entrainment of stratospheric $O_3$, a key source of tropospheric $O_3$ in the pre-industrial atmosphere, particularly at the poles. Previous studies have demonstrated that elevated $O_3$ levels in the Arctic during MAM and DJF, as well as in the Antarctic during JJA and SON, result from the cumulative impact of the polar $O_3$ barrier (Romanowsky et al., 2019).

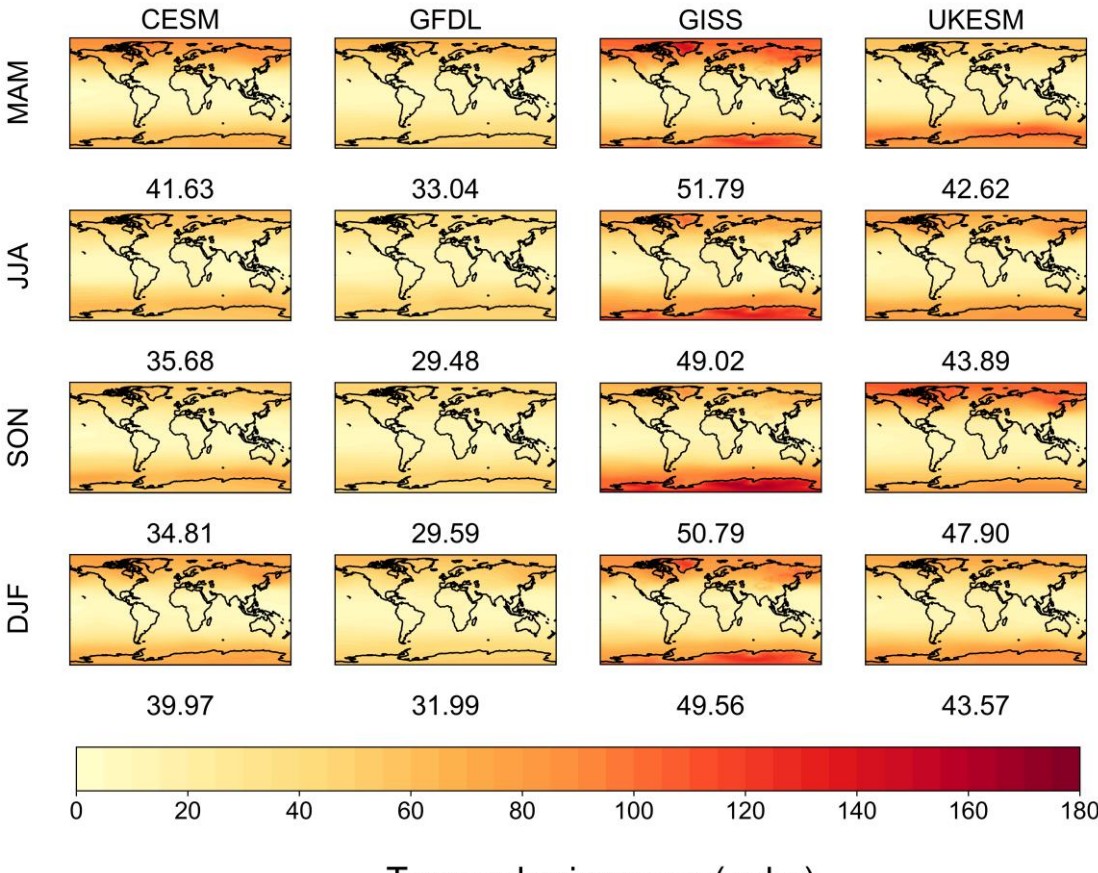

**Figure 1.** Comparison of the seasonal cycle of tropospheric column averaged volume mixing ratio of $O_3$ (density weighted) of the *PiClim* experiment results in the 29[th] and 30[th] year of simulation of the four models. Each row shows a separate meteorological season, arranged from top to bottom: March to May (MAM), June to August (JJA),

September to November (SON), and December to February (DJF). Each column
represents a selected model, listed from left to right: CESM, GFDL, GISS, and
UKESM. The figures displayed below each chart represent the global average ozone
volume mixing ratio.

Seasonal variations in tropospheric $O_3$ volume mixing ratio exhibit model-
specific patterns. The CESM, GFDL, and GISS models simulate peak tropospheric
$O_3$ volume mixing ratio in spring during the *PiClim* experiments. In contrast, the
UKESM model reproduces maximum $O_3$ volume mixing ratio in autumn, indicating
a limited capability in simulating dynamic circulations in the tropopause.
Furthermore, the seasonal $O_3$ cycle simulations in CESM, GFDL, and GISS exhibit
distinct discrepancies in their outcomes. For instance, the CESM model simulates the
lowest $O_3$ volume mixing ratio in SON, while the GFDL model exhibits the lowest
volume mixing ratio in JJA. The GISS model simulation indicates higher $O_3$ levels in
autumn compared to DJF, which is consistent with results from historical experiments
(Griffiths et al., 2021). Additionally, our analysis reveals that the CESM simulations
demonstrate the most pronounced seasonal oscillation amplitude in $O_3$ volume
mixing ratio, approximately 6.82 ppbv. This feature underscores the model's
sensitivity to seasonal factors affecting tropospheric $O_3$ dynamics.

In the *PiClim* experiments, all four models accurately reproduce the peak volume
mixing ratio of $O_3$ in the middle stratosphere at 10 hPa and the zonal average mixing
ratios reaching their peak in the upper troposphere, particularly in extratropical
regions, indicative of extended chemical lifetimes at higher altitudes. However,
notable disparities are observed in the vertical distribution characteristics of $O_3$
among the four models (Fig. 2). Specifically, the CESM model exhibits the highest
vertical extension, including an additional hotspot simulated in the thermosphere.
While the GFDL and CESM2 models exhibit consistent simulation outcomes below
0.01 hPa, GISS and UKESM simulate significantly higher stratospheric $O_3$ levels at
10 hPa in comparison.

Notable distinctions are observed in the spatial distribution of $O_3$. The GISS
model simulates a more vertically concentrated and latitudinally extended $O_3$
distribution. This characteristic may be a crucial factor contributing to the pronounced
impact of $O_3$ transport in the polar stratosphere, as simulated by GISS. The zonal
variability in $O_3$ distribution simulated by the UKESM falls between that of the GISS
and CESM models. These inter-model discrepancies in $O_3$ simulation results likely
reflect suboptimal representation of local and regional dynamics, as well as omitted
chemical processes in corresponding models. The variability and uncertainty in $O_3$
precursor emission estimates further exacerbate these disparities.

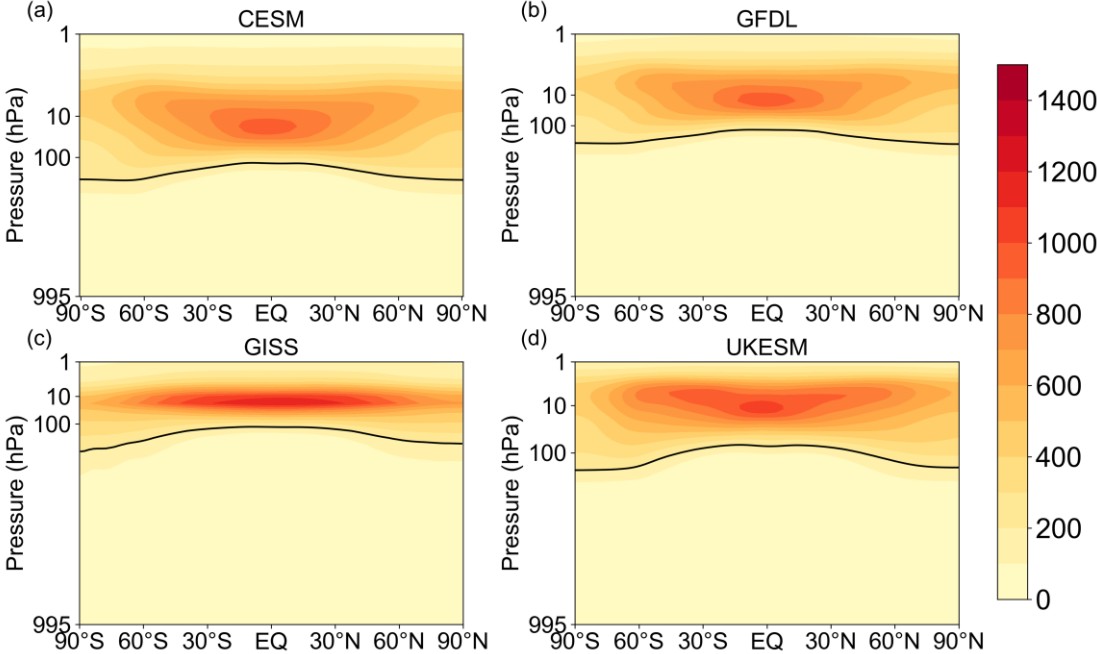

**Figure 2.** The zonal mean $O_3$ distribution for the $29^{th}$ and $30^{th}$ year of the *PiClim*
experiment results from the (a) CESM, (b) GFDL, (c) GISS, and (d) UKESM model.
Thick black lines represent the tropopause height for each model based on the WMO
definition.
**3.2 Characteristics of tropospheric $O_3$ under various experiments**
Tables 3 and 4 present the global $O_3$ volume mixing ratio and tropospheric $O_3$
volume mixing ratio across all experiments from the four different models. The GISS
model simulations show higher tropospheric $O_3$ volume mixing ratios, reflecting
increased rates of stratospheric downwelling and surface $O_3$ precursor emissions.
However, its overall $O_3$ volume mixing ratio is notably lower compared to the
UKESM, CESM, and GFDL models, with reductions of 114.24, 76.16, and 47.04
ppbv, respectively. Analysis reveals that in the CESM, GFDL, and GISS models, the
global $O_3$ molar fraction in the *PiClim-2NOx* and *PiClim-NOx* experiments surpasses
that in the *PiClim-2VOC* and *PiClim-VOC* experiments. This difference is most
pronounced in the GISS model, aligning with previous findings indicating its
heightened sensitivity to $NO_x$ response (Turnock et al., 2019). Conversely, in the
UKESM model, the global $O_3$ molar fraction of the *PiClim-2NOx* experiment is lower
than that of the *PiClim-2VOC* experiment. Interestingly, the tropospheric $O_3$
volume mixing ratios in the *PiClim-2NOx* experiment in the CESM and GFDL models
are notably lower than in their respective *PiClim-2VOC* experiments, with reductions
of 0.41 and 0.29 ppbv. This discrepancy challenges the conventional understanding
that increased $NO_x$ emissions from lightning activity should lead to tropospheric $O_3$
generation, suggesting a need for enhanced sensitivity simulations in these two
models regarding $O_3$ and $NO_x$ emissions from natural sources due to lightning activity.
In contrast, the *PiClim-2NOx* experiments of the GISS and UKESM models
effectively simulate an increase in tropospheric $O_3$ volume mixing ratio compared to
their *PiClim-2VOC* experiments. Furthermore, across all four models, the
tropospheric $O_3$ volume mixing ratio of the *PiClim-NOx* experiment surpasses that of
the *PiClim-VOC* experiment, indicating the models' ability to accurately replicate the
impact of rising anthropogenic emissions on $O_3$ production. Additionally, methane, a
crucial natural source of volatile organic compounds and a key greenhouse gas,
enhances tropospheric $O_3$ generation by $CH_4$ oxidation and influencing temperature,
thereby elevating global $O_3$ volume mixing ratio. This phenomenon contributes to the
heightened sensitivity of $O_3$ to methane volume mixing ratio in a clean atmosphere.
Elevated volume mixing ratios of HCFCs (*PiClim-HC*) and nitrous oxide (*PiClim-*
*N2O*) lead to substantial stratospheric $O_3$ depletion, consequently affecting
tropospheric $O_3$ volume mixing ratio through the pod coil process. Other influencing
factors, such as aerosols and black carbon, induce warming through radiation effects,
thereby simulating elevated $O_3$ volume mixing ratio.

Table 3. The averaged volume mixing ratio of global stratospheric ozone at all simulated vertical levels in the 29th and 30th year for each experiment of four models (ppbv).

| Model \ PiClim- | $2xNO_x$ | $2xVOC$ | HC | $CH_4$ | $NO_x$ | VOC | NTCF | $N_2O$ | $O_3$ | aer | control | BC |
|---|---|---|---|---|---|---|---|---|---|---|---|---|
| CESM2-WACCM | 726.06 | 725.95 | 662.71 | 713.80 | 728.61 | 727.06 | 725.42 | 710.94 | | | | |
| GFDL-ESM4 | 628.63 | 626.68 | 571.32 | | 632.03 | 628.92 | | | 632.70 | 628.44 | 629.98 | 629.78 |
| GISS-E2-1-G | 490.91 | 482.13 | 422.65 | 493.27 | 490.22 | 480.93 | 486.46 | 471.84 | 485.17 | 486.54 | 484.76 | 484.82 |
| UKESM1-0-LL | 707.27 | 707.93 | 613.89 | 697.32 | 716.14 | 704.78 | 723.99 | 694.44 | 714.27 | 697.04 | 702.88 | 701.81 |

**Table 4.** The averaged volume mixing ratio of global tropospheric ozone in the 29[th] and 30[th] year for each experiment of four models (ppbv).

| Model \ PiClim- | 2xNO$_x$ | 2xVOC | HC | CH$_4$ | NO$_x$ | VOC | NTCF | N$_2$O | O$_3$ | aer | control | BC |
|---|---|---|---|---|---|---|---|---|---|---|---|---|
| CESM2-WACCM | 38.17 | 38.58 | 33.44 | 39.42 | 39.16 | 39.14 | 41.33 | 38.10 | | | | |
| GFDL-ESM4 | 31.33 | 31.62 | 24.42 | | 32.64 | 32.25 | | | 34.09 | 31.01 | 30.79 | 30.95 |
| GISS-E2-1-G | 52.30 | 50.96 | 44.18 | 53.08 | 52.14 | 50.21 | 51.65 | 48.36 | 52.47 | 50.36 | 49.27 | 50.02 |
| UKESM1-0-LL | 47.53 | 46.14 | 31.04 | 45.55 | 46.02 | 45.97 | 47.29 | 45.04 | 46.65 | 43.69 | 46.70 | 45.11 |


Figure 3 shows the temporal evolution of tropospheric $O_3$ levels across various
latitudes, as simulated by four distinct models in $O_3$ precursor experiments. In the
*PiClim* experiments, none of the models predicted an enhancement in $O_3$ volume
mixing ratio with simulation time at all latitudes, reflecting the consistent chemical
lifetime of $O_3$ within the pristine atmospheric conditions. However, discrepancies in
$O_3$ predictions among the models become more pronounced with increasing latitudes.
While the CESM model generally exhibits higher tropospheric $O_3$ volume mixing
ratios compared to the GFDL model, it paradoxically portrays the lowest $O_3$ levels in
the equatorial region. The GISS model demonstrates a marked disparity in
tropospheric $O_3$ volume mixing ratios between the Antarctic and Arctic regions, with
the former registering notably higher levels. In contrast, the CESM and GFDL models
exhibit similar patterns in this regard. A unique feature of the GISS model is a notable
declining trend in Antarctic tropospheric $O_3$ levels during the initial 15 years of both
the *PiClim-2VOC* and *PiClim-VOC* experiments. This trend is not observed in the
CESM, GFDL, and UKESM models, highlightingthe sensitivity of the GISS model
to precursors in simulating ozone is still higher than that of other models even in the
pre-industrial clean atmosphere. The same conclusion was reached for $NO_x$
experiments, but the ozone forcing was less than that in the VOC experiments. The
UKESM model stands out with its pronounced simulation of elevated $O_3$ volume
mixing ratios in the tropical belt. Furthermore, the *PiClim-2xVOC* experiment
conducted within the UKESM model demonstrates a significant $O_3$ response to
enhanced emissions of VOCs from natural sources in the equatorial region. This
suggests a strong sensitivity of $O_3$ in the UKESM to increases in VOC emissions from
natural sources.

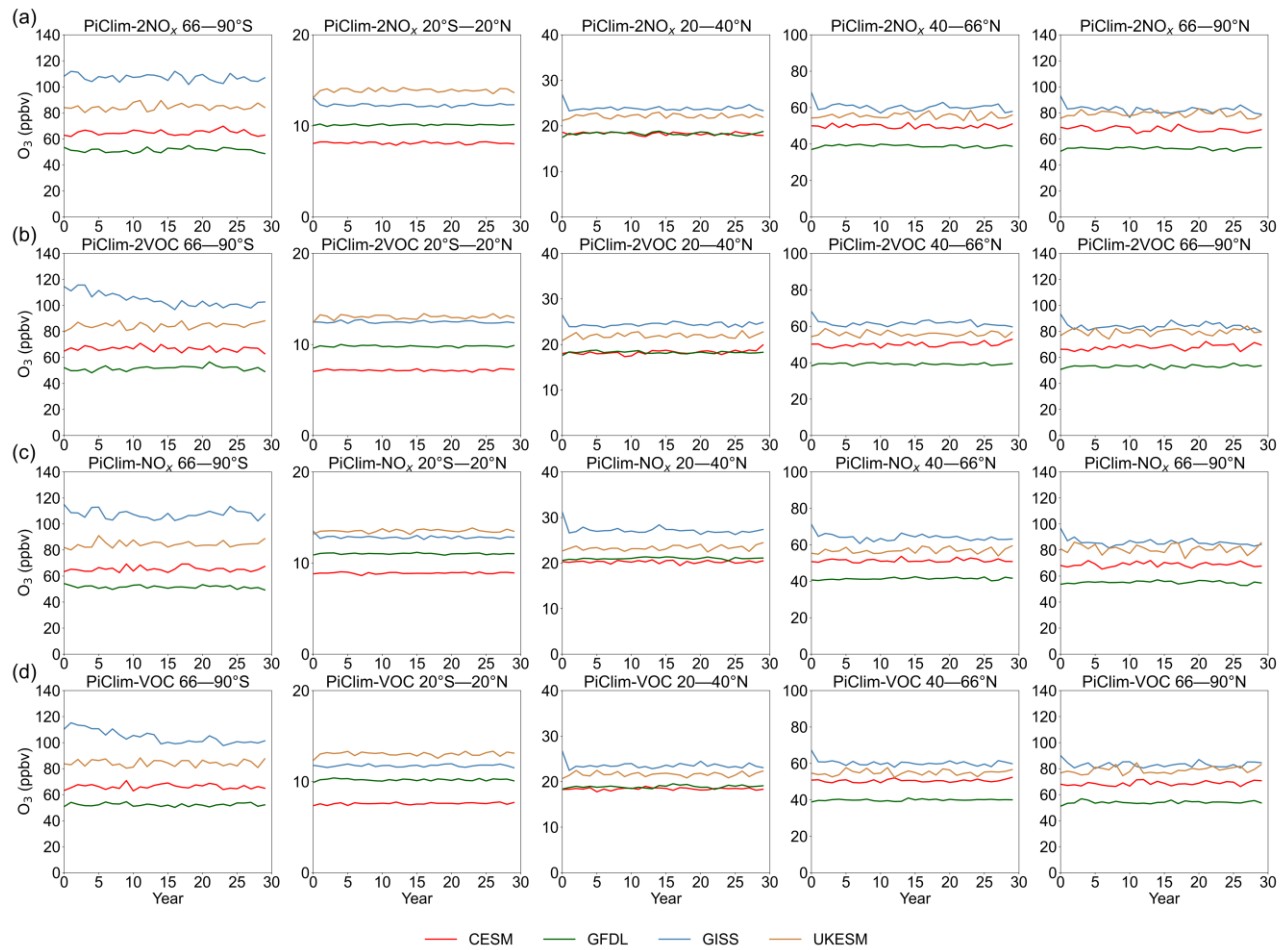

**Figure 3.** The temporal evolution characteristics of annual mean tropospheric column averaged $O_3$ volume mixing ratio at different latitudes for each model are presented for the (a) *PiClim-2NOx*, (b) *PiClim-2VOC*, (c) *PiClim-NOx*, and (d) *PiClim-VOC* experiment, the 4 models are represented by different line colors.

**3.3 Analysis of O₃ generation in precursor experiments**

In the shown subset of *PiClim* experiments, the O₃ production was defined as the cumulative tendency from $HO_2$, $CH_3O_2$, $RO_2$, and NO reactions, while O₃ loss encompassed the sum of $O(1D) + H_2O$, $O_3 + HO_2$, $OH + O_3$, and $O_3$ + alkene reactions. Figure 4 depicts the chemical production and consumption of tropospheric ozone in the five simulations performed by the four models. The GISS demonstrates the lowest O₃ chemical production among the models, whereas the other three models show generally consistent production levels. Notably, the GISS model exhibits a relatively low efficiency in O₃ chemical consumptions, primarily due to missing the loss of O₃ with isoprene and terpenes process. The low offset of ozone production and depletion in the pre-industrial atmosphere by the GISS model provides a new perspective based on previous studies indicating the high offset of ozone production and depletion in the present atmosphere by the GISS model. The four models all showed high ozone chemical production in the *PiClim-NOₓ* experiment, indicating that the four all have perfect ability to simulate the photochemical generation mechanism of tropospheric ozone. However, the CESM and GFDL models do not show a significant increase in tropospheric O₃ chemical generation during the *PiClim-2NOₓ* experiment. And although the GISS and UKESM models successfully simulated an increase in the O₃ chemical generation rate due to heightened lightning activity in this experiment, these increases in ozone production are also much smaller than the chemical production generated by the *PiClim-NOₓ* experiment, which might show that the theoretical mechanism of ozone sensitivity to natural precursors in pre-industrial atmosphere differs from the present mechanism due to the differences in the characteristics of intermediate products such as OH. Furthermore, in either model, the ozone chemical production from the *PiClim-NOₓ* experiment, while higher than in other experiments other than *PiClim-NTCF*, is much smaller than the ozone chemical production caused by this emission inventory in the atmosphere today (Fig. S5). Today's NOₓ emission forcing has not led to a sustained increase in the ozone volume mixing ratio in the pre-industrial atmosphere over a long-time scale, which indicates important differences between the pre-industrial atmosphere and the present atmosphere in terms of the ozone generation environment and the ozone depletion environment.

Furthermore, the *PiClim-2VOC* experiment in the CESM and GFDL models lead to an increase in tropospheric O₃ volume mixing ratio, despite not reproducing higher

$O_3$ chemical production. The UKESM model successfully captures the enhancement
of $O_3$ chemical formation due to increased emissions of VOCs from natural sources,
underscoring its precise sensitivity to these emissions and validating its capability to
simulate $O_3$ dynamics influenced by them. However, the global $O_3$ volume mixing
ratio in the *PiClim-2xVOC* experiment of these models is lower than that of the
*PiClim-VOC* experiment. These observations illustrate the variability among models
in capturing the $O_3$ response to its precursor species, stemming from varied treatments
of critical atmospheric processes, including photolysis, dry deposition, transport
mechanisms, and mixing dynamics. Furthermore, these findings highlight the
variability in global $O_3$ sensitivity compared to local $O_3$ sensitivity, underscoring the
complexity of studying $O_3$ sensitivity on a global scale to mitigate its climate impacts.

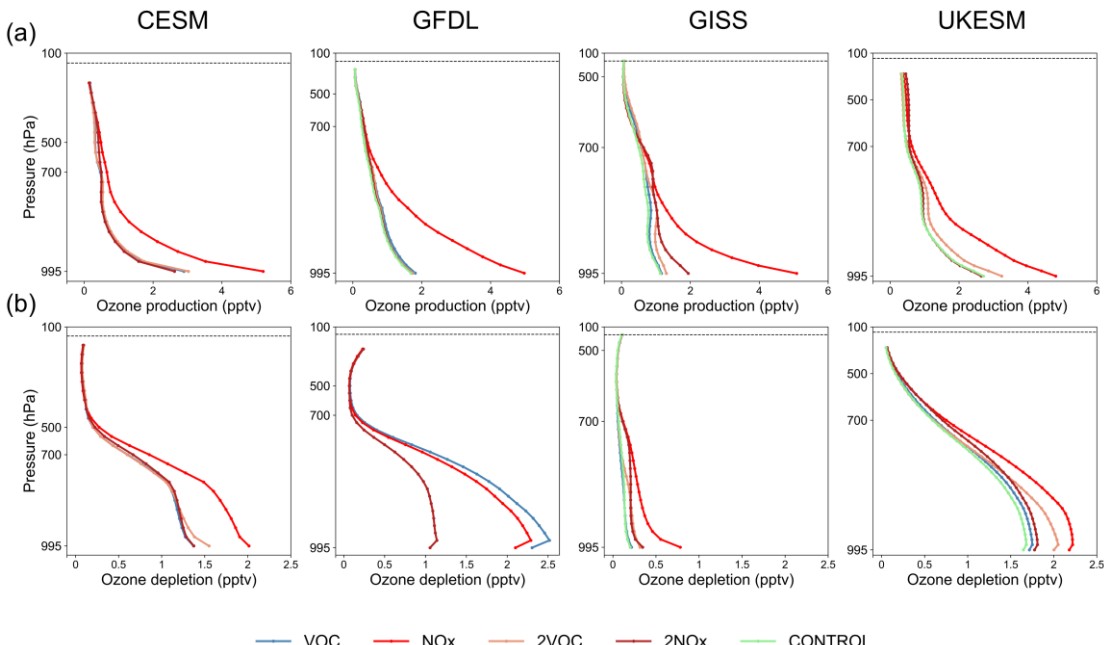


**Figure 4.** Vertical profiles of $O_3$ volume mixing ratio (a) chemical production and (b)
chemical depletion rate for the 30[th] year across five experiments in the four models.
Figure 4b illustrates that, apart from the $O_3$ chemical formation mechanism, the
CESM, GFDL, and UKESM models in the *PiClim-2NOx* experiment do not
accurately depict the $O_3$ chemical depletion process induced by $NO_x$. Despite
successfully replicating the rise in NO and $NO_2$ levels (Fig. 5a, b) in the upper
troposphere, these models fall short in capturing the $NO_x$-related $O_3$ depletion
phenomenon. Moreover, the GISS model stands out with notably elevated $NO_x$
volume mixing ratios attributed to heightened lightning activity compared to the other
models. Additionally, it demonstrates a peak $NO_x$ volume mixing ratio near 500 hPa
across these four experiments conducted, a feature not observed in the other models.

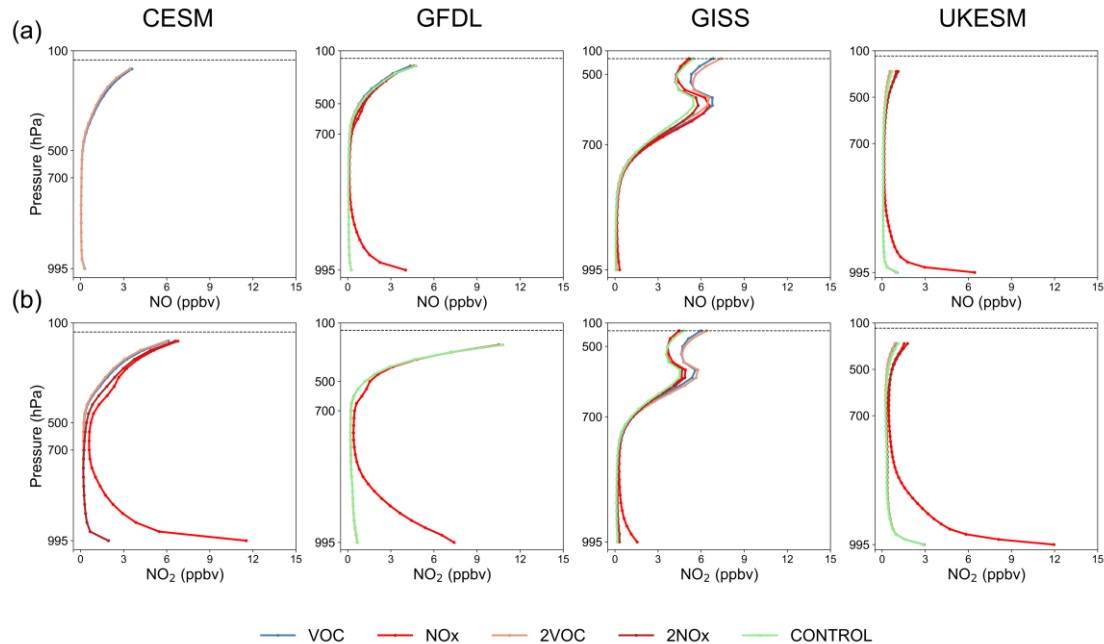

**Figure 5.** Vertical profiles of (a) NO and (b) $NO_2$ volume mixing ratios for the 30th year across five experiments in the four models.

Figure 6 illustrates a notable inverse correlation between the consumption of isoprene and the chemical production of $O_3$ in four models, when the rise in VOCs emissions is not factored in. This relationship is attributed to the significance of isoprene as a natural VOC source in unpolluted atmospheres and highlights the absence of $O_3$ generation simulation due to lightning activity in the CESM, GFDL, and UKESM models. In the *PiClim* experiments, the UKESM model did not provide mass fraction of secondary particulate organic matter dry aerosol particles in the air (mmrsoa), and so we only include its volume mixing ratio of isoprene in the air (isop) and the primary emissions and chemical production of dry aerosol organic matter (emioa) in Fig. 6. Additionally, the CESM model exhibits higher emissions and chemical formation of organic dry aerosol particles compared to the GFDL and GISS models. This difference potentially contributes to the observed variation in global $O_3$ volume mixing ratios, with the highest levels recorded in the CESM model and the lowest in the GISS model.

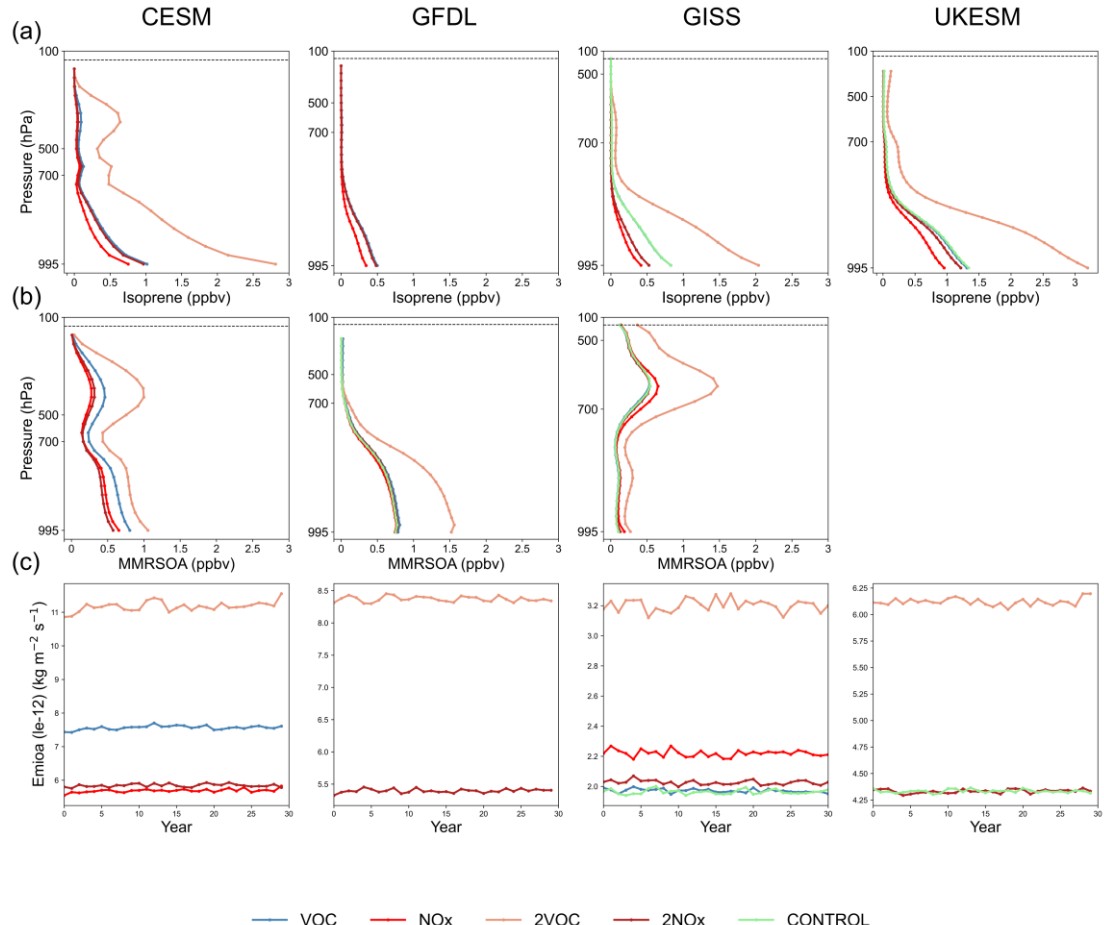

**Figure 6.** Vertical profiles of (a) isoprene volume mixing ratio and (b) secondary organic aerosol mass mixing ratio for the 30th year of all available experiments across the three models. (c) Temporal evolution characteristics of major emissions and the chemical production of organic dry aerosol particles from five experiments of the four models.

## 4. Conclusions

This study assessed the sensitivity of global-scale ozone ($O_3$) to precursor gases in a clean atmosphere and evaluated the simulation capabilities of four Earth system models using data from the *PiClim* experiments within the AerChemMIP framework. Our results highlight both strengths and limitations of these models in capturing $O_3$ response. The CESM and GFDL models excelled in reproducing seasonal $O_3$ cycles and the vertical distribution of $O_3$, but they showed limitations in simulating the tropospheric $O_3$ response to $NO_x$ emissions from natural sources, such as lightning activity. Conversely, the GISS and UKESM models effectively simulated the positive correlation between tropospheric $O_3$ and temperature but were less sensitive to natural precursors compared to anthropogenic sources. Discrepancies, such as zonal

temperature biases in the GISS model and stratospheric temperature inconsistencies
in the GFDL model, underscore areas for improvement.
Our findings suggest that existing assumptions regarding $O_3$ sensitivity to
natural precursors may require refinement in clean atmospheric conditions. This
research provides critical insights into the interplay between $O_3$ and its precursors,
enhancing the accuracy of $O_3$ simulations in Earth system models. Given the
significant role of $O_3$ in radiative forcing, atmospheric oxidation, and climate
feedback mechanisms, our study reinforces the necessity of precise modeling to better
predict and mitigate future climate scenarios. Additionally, the results underscore the
importance of controlling anthropogenic precursor emissions as an essential strategy
to manage tropospheric $O_3$ volume mixing ratios and address broader climate change
challenges. Furthermore, among the models analyzed, only the GISS model
demonstrates a significant increase in Antarctic ozone levels compared to the Arctic
(Fig. 3); the other three models yield similar ozone concentrations at both polar
regions. This discrepancy seems to result from a distinct characteristic of the GISS
model's dynamical representation of the Antarctic polar vortex. Figure 1 also reveals
that the ozone difference in the GISS model is predominantly confined to JJA and
SON (Antarctic winter-spring).
It is important to acknowledge that the results generated by the models are
accompanied by a degree of uncertainty. Variations in the methodologies employed
by different models to address chemical reactions, including the production and
depletion of ozone, contribute to the uncertainty surrounding the ozone budget.
Furthermore, discrepancies in the data pertaining to anthropogenic and natural
emissions, particularly concerning $NO_x$ and BVOC emissions, substantially influence
the outcomes of these models. Additionally, the uncertainty associated with the
stratosphere-troposphere exchange process represents a critical factor in the ozone
budget, with notable divergences in the treatment of this process across various
models.
**Acknowledgement**
We acknowledge the World Climate Research Programme, which, through its
Working Group on Coupled Modelling, coordinated and promoted CMIP6. We thank
the climate modelling groups for producing and making available their model output,

the Earth System Grid Federation (ESGF) for archiving the data and providing access, and the multiple funding agencies who support CMIP6 and ESGF. We acknowledge the AerChemMIP groups of the four models used in the study (Vaishali Naik and Larry Horowitz for the GFDL simulations, Susanne E. Bauer and Kostas Tsigaridis for the GISS simulations, Fiona O'Connor and Jonny Williams for the UKESM simulations, as well as Louisa K. Emmons for the NCAR simulations). Particularly, we are grateful to Dr. Vaishali Naik for her comments and suggestions during the revision of this manuscript. We also thank the editor and anonymous reviewers for their time and comments, which helped improve the quality of this work greatly.

**Data availability**

All data from the Earth system models used in this paper are available on the Earth System Grid Federation website and can be downloaded from https://esgf-index1.ceda.ac.uk/search/cmip6-ceda/ (last access: 4 July 2024, ESGF-CEDA, 2024).

**Author contributions**

WW and CYG provided data analysis and contributed to the writing and discussion of this paper.

**Competing interests**

The authors declare that they have no conflict of interest.

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
