# Peer review of "Global Sensitivity of Tropospheric Ozone to Precursor Emissions in"

_EGUsphere, 2024_

## Author Comment (AC2)

Dear Editor and Reviewers,

Thank you for your thoughtful review of our manuscript and for providing constructive feedback that has strengthened our work.

We have carefully addressed all comments and made substantial revisions to the manuscript. The major changes include comprehensive revisions to Tables 1-4 and Figures 2-6, which directly address reviewers' primary concerns regarding visual presentation quality.

Our detailed point-by-point responses to each reviewer's comments are provided below, with our responses in blue text. All corresponding changes in the manuscript are highlighted in blue to facilitate your review.

We believe these revisions have significantly improved the manuscript and look forward to your consideration.

Sincerely,

The Authors

**Report #1 Anonymous Referee #2**
**General comments**

The paper presents the sensitivity of ozone on precursor gases, anthropogenic ozone depleting gases and aerosol in the framework of a comparison of chemistry climate models. The main focus is on tropospheric ozone but results are presented only in crude integral quantities where effects often cancel out. One figure shows that the noise due to meteorology appears to be dominating in the 'time-slice' simulations (?) where correlations between ozone and temperatures even change sign arbitrarily. There is some lack of significance, especially if only the arbitrary values of the last simulation year are picked because of model dependent interannual variability like e.g. the QBO (Quasi-Biennial Oscillation) which is even not mentioned in the text. A problem in this paper is that the figures are in a poor quality concerning definition of the shown quantity, unclear units, ignoring conventions for axes and too many lines in a frame which cannot be distinguished (Figs. 5 - 8). The worst of this is Fig. 2 with arbitrary z-axes (model levels?). Please use here a logarithmic pressure axis or altitude and more ticks.
The conclusions cannot be drawn from the presented figures and should be expanded.

**Specific comments**

Line 18: In preindustrial atmospheres OH is not globally uniform due to seasonal variation, a latitude dependence, land sea differences, convection and land cover. This sentence is misleading and has to be improved.

Thank you for pointing this out, it is in fact an oversight, and we have revise the text to be more accurate. The text now reads, "*This study investigates the sensitivity of global $O_3$ to precursor gases in a clean atmosphere, where hydroxyl (OH) radical characteristics are more spatially uniform than in present-day conditions, using data from the PiClim experiments of the Aerosols and Chemistry Model Intercomparison Project (AerChemMIP) within the CMIP6 framework.*"

Line 26: Surface temperature? Or some tropospheric average?

Upon considerations, we have removed it from the text to makes the abstract more concise and focused on the key findings.

Line 29: Do you mean here "radiative forcing"? Or just photochemical smog reactions?

Thank you for asking, we realize the original text was ambiguous and have revised the text to make it more clear. The text now reads, *"While all models successfully simulate O3 responses to anthropogenic precursor emissions, CESM and GFDL show limited sensitivity to enhanced natural NOx emissions (e.g., from lightning) compared to GISS and UKESM."*

Line 30ff: What is the main result? Just noise? Please be more specific.

We have expanded the sentence and now it reads, "The sensitivities of O3 to its natural precursors (NOx and VOCs) in GISS and UKESM models are substantially lower than their responses to anthropogenic emissions, particularly for lightning $NO_x$ sources."

Line 44 or later: OH is also sensitive to absolute humidity (i.e. temperature dependent for fixed relative humidity) and CO.

Thank you for your suggestion, we have considered OH's physical and chemical properties, and while OH is sensitive to humidity and CO, it is not the focus of this study, so we have decided to keep the statement in Line 44 as is.

Line 56ff: I miss some key references and mentioning 'NOx-limited' and 'VOC-limited'.

We have added three references "*Regional-scale sensitivity to $O_3$ precursors has been extensively investigated, such as emphasizing the diagnostic utility of ratios including $O_3/NO_x$ (Jin et al., 2023; Sillman and He, 2002) and $VOC/NO_x$ (Li et al., 2024) for assessing $O_3$-$NO_x$-VOC sensitivity...*"

Table 1:   Contains 1 useless column (content can be in caption or text) but also definitions are missing (number of gridpoints, Eulerian or spectral model?). Here other important properties, e.g. chemistry module or boundary conditions (BVOC) might be included.

Thanks for your suggestions, the models are all Eulerian. We have streamlined Table 1 by removing redundant information and make it clear, and information about chemistry module and BVOC are included in the model description so as to not overcrowd the table.

**Table 1.** Information on model resolution, vertical levels, property of gas-phase chemistry and references.

| Model | Resolution (lat × lon) | Vertical levels | Tropospheric and stratospheric chemistry | Aerosol model | Simulation reference |
|---|---|---|---|---|---|
| CESM2-WACCM | 192 × 288 | 70 levels; top level 6 × 10-6 hPa | | MAM4 | (Gettelman et al., 2019) |
| GFDL-ESM4 | 180 × 288 | 49 levels; top level 0.01 hPa | | MATRIX | (Dunne et al., 2020; Horowitz et al., 2020) |
| GISS-E2-1-G | 90 × 144 | 40 levels; top level 0.1 hPa | Interactive | OMA | (Miller et al., 2014; Kelley et al., 2020) |

| UKESM1-0-LL | 144 × 192 | 85 levels; top level 1 hPa | | GLOMAP | (Mulcahy et al., 2018; Sellar et al., 2019) |

Line 110 or later (line 240?): Important information concerning emissions and boundary conditions for gases like CO and CH$_4$ is missing, as well as a table containing emission inventories (references) for the used gases and aerosol particles. Some information on a subset of species is scattered over the text.

Thanks for raising this concern, emissions used clean atmosphere emission inventory, and we think rather than repeating information in a table, information included in texts where they are discussed would make our message more clear.

Line 186: Right wording? Do you use CO and HCHO as intermediate products of NMVOC oxidation?

Yes. CO and HCHO are treated according to each model's default chemistry scheme, which includes both direct emissions and formation as NMVOC oxidation products.

Line 189: WMO, dynamical or O$_3$? Mention what is used, if models are different here it should be included in Table 1.

The definition was mentioned in Figure 2's caption, and we have included in the text now too, "*All four of the interactive tropospheric chemistry models contain parameterizations of the nitrogen oxide (NO$_x$) emissions from lightning based on the height of the convective cloud top (Price et al., 1997; Price and Rind, 1992; Price, 2013), and the tropopause height for each model based on the WMO definition.*"

Line 194: This should also go into Table 1.

Thanks for the suggestion, however, we think too much information in Table 1 would overcrowd the table, so we have decided to only keep the essential information and the table has been updated to include more information, as shown above.

Line 215: Are these free running 'time-slice'-simulations with fixed boundary conditions for about the year 1850 without and with perturbations?

Yes, these are free-running time-slice simulations with boundary conditions representative of 1850 (pre-industrial) conditions. The PiClim-control represents unperturbed pre-industrial conditions, while other PiClim experiments apply specific perturbations (e.g., doubled natural emissions in PiClim-2x experiments, present-day emission levels in PiClim-x experiments) to this baseline state.

Line 221ff: It would be good to have some list or table in an Appendix with what is included in VOC, BVOC, aer, HC and NTCF. The assumptions in the scenarios are rather restricted, do you e.g. assume that soil and forest fire sources for NOx stay unchanged?

Thanks for your suggestions. The components of each scenario are described in Section 2.2 (lines 238-248). Regarding soil and forest fire NOx sources, these natural emission sources remain at pre-industrial levels in all experiments, with only the specific targeted emissions (anthropogenic NOx, VOCs, etc.) being perturbed according to each scenario design.

Line 227: Inconsistent with line 400. Halocarbons include CFCs and HCFCs but also compounds containing bromine. $CH_3Cl$, $CH_3Br$ and $CCl_4$ are the most important with natural emissions. What is used? I hope that for increased CFCs consistent initial conditions with upscaling are used for ClY and BrY concentrations to prevent a drift lasting decades.

sign.Thank you for pointing it out. We have already corrected it to "halocarbons include CFCs, HCFCs and compounds containing bromine". And in the PiClim experiments, all initial conditions were based on the 1850 emissions inventory, and the diffusion scales of these substances were consistent.

Table 2: Include a row in the header with main features of the scenarios.

Thanks for the suggestion, Table 2 has been updated:

**Table 2.** The available experiments of selected models in this study. "X" represents the experiment is available.

| Model | piClim-2xNO$_x$ | piClim-2xVOC | piClim-HC | piClim-CH$_4$ | piClim-NO$_x$ | piClim-VOC | piClim-NTCF | piClim-N$_2$O | piClim-O$_3$ | piClim-aer | piClim-control | piClim-BC |
|---|---|---|---|---|---|---|---|---|---|---|---|---|
| CESM2-WACCM | X | X | X | X | X | X | X | X | | | | |
| GFDL-ESM4 | X | X | X | | X | X | | | X | X | X | X |
| GISS-E2-1-G | X | X | X | X | X | X | X | X | X | X | X | X |
| UKESM1-0-LL | X | X | X | X | X | X | X | X | X | X | X | X |

Line 240: Emission data here? Or in line 243? Please more details.
Line 240 provides variable name definitions. Emission data details are provided earlier in lines 233-237, with specific inventory references (CEDS: Hoesly et al., 2018; biomass burning: van Marle et al., 2017).

Line 266f: The numbers would be different for the 29[th] year. Use at least a 2 year average because of QBO. This holds for all results shown later. For the greenhouse forcing the upper and mid tropospheric ozone matters more   than the surface one.

Thank you for pointing it out! We have changed the research on ozone concentration involved in Figures 1-2 and Tables 3-4 to two-year average values to avoid the influence of oscillation.

Line 276: Does this refer to troposphere or stratosphere? Austral summer? Known is the Antarctic vortex in the lower stratosphere in Austral winter as transport barrier and its relation to the present day ozone hole. Please be more precise here. This is confusing as well as the caption of Fig. 1 where the season names appear to refer to the Northern hemisphere.

Thanks for bringing this to our attention, we have changed seasons to monthly abbreviations throughout the manuscript, e.g. *"Previous studies have demonstrated that elevated $O_3$ levels in the Arctic during MAM and DJF, as well as in the Antarctic during JJA and SON, result from the cumulative impact of the polar $O_3$ barrier (Hamlin and Honrath, 2002)." "For instance, the CESM model simulates the lowest $O_3$ volume mixing ratio in SON, while the GFDL model exhibits the lowest volume mixing ratio in JJA. The GISS model simulation indicates higher $O_3$ levels in autumn compared to DJF, which is consistent with results from historical experiments (Griffiths et al., 2021)."*

Line 312ff: This cannot be seen from Fig. 2 which is distorted for each model in a different way (see also general remarks). Here it might be useful to show DJF and JJA seasons instead of the annual average.

We have revised Figure 2 with individual axes for each model (no longer compressed with uniform scales) and added separate panels showing both annual average and individual seasons (DJF, MAM, JJA, SON in Supplementary Information) as suggested.

[Figure]

Fig.3: I suppose you mean density weighted vertical average with variable tropopause and stratopause? Please specify in caption. Better use fixed levels like 100 and 1hPa because of the radiative heating characteristics. Surfacetro? This is not a common meteorological word. Do you mean temperature of the surface layer? For scientific interpretation this figure is of rather limited value, due to compensating effects in different altitude levels from chemistry and radiation.

Thanks for bringing this to our attention, to avoid confusion, we have deleted this figure.

Fig. 4: Looks like noise due to meteorological variation in different altitudes. It is known from textbooks and the IPCC reports that the correlation of $O_3$ in different altitudes with surface temperature can change sign. Please redesign or skip this figure.

Thanks, we have taken your suggestion to skip this figure.

Table 3: Please provide this for the stratosphere or better the lower stratosphere (below 10hPa). The total average numbers are almost useless because of compensating effects. For surface climate only the lower stratosphere matters. Here also some sentences on the Antarctic and Arctic ozone hole formation and model differences in the HC-scenario would useful. Provide proper definition of what is shown in the caption (or refer at least to figure captions or an expanded Table 2).

Thanks for your suggestions, Table 3 has been updated:

**Table 3.** The averaged concentrations of global ozone at all simulated vertical levels in the 30th year for each experiment of four models (ppbv).

| Model | piClim -2xNO$_x$ | piClim -2xVOC | piClim -HC | piClim -CH$_4$ | piClim -NO$_x$ | piClim -VOC | piClim -NTCF | piClim -N$_2$O | piClim -O$_3$ | piClim -aer | piClim -control | piClim -BC |
|---|---|---|---|---|---|---|---|---|---|---|---|---|
| CESM2-WACCM | 398.62 | 398.56 | 363.84 | 391.89 | 400.20 | 399.17 | 398.27 | 390.32 | | | | |
| GFDL-ESM4 | 365.48 | 364.35 | 332.16 | | 367.46 | 365.65 | | | 367.85 | 365.37 | 366.27 | 366.15 |
| GISS-E2-1-G | 322.97 | 317.19 | 278.06 | 324.52 | 322.51 | 316.40 | 320.04 | 310.42 | 319.19 | 320.09 | 318.92 | 318.96 |
| UKESM1-0-LL | 435.24 | 435.65 | 377.78 | 429.12 | 440.70 | 433.71 | 445.53 | 427.35 | 439.55 | 428.95 | 432.54 | 431.88 |

Table 4: Expand caption as above.

Thanks for your suggestions, Table 4 has been updated:

**Table 4.** The averaged concentrations of global tropospheric ozone in the 30th year for each experiment of four models (ppbv).

| Model | piClim-2xNO$_x$ | piClim-2xVOC | piClim-HC | piClim-CH$_4$ | piClim-NO$_x$ | piClim-VOC | piClim-NTCF | piClim-N$_2$O | piClim-O$_3$ | piClim-aer | piClim-control | piClim-BC |
|---|---|---|---|---|---|---|---|---|---|---|---|---|
| CESM2-WACCM | 38.17 | 38.58 | 33.44 | 39.42 | 39.16 | 39.14 | 41.33 | 38.10 | | | | |
| GFDL-ESM4 | 31.33 | 31.62 | 24.42 | | 32.64 | 32.25 | | | 34.09 | 31.01 | 30.79 | 30.95 |
| GISS-E2-1-G | 52.30 | 50.96 | 44.18 | 53.08 | 52.14 | 50.21 | 51.65 | 48.36 | 52.47 | 50.36 | 49.27 | 50.02 |
| UKESM1-0-LL | 47.53 | 46.14 | 31.04 | 45.55 | 46.02 | 45.97 | 47.29 | 45.04 | 46.65 | 43.69 | 46.70 | 45.11 |

Line 401: Isn't it PiClim-N$_2$O?

Thanks for pointing it out. This was a typo and has been corrected to "methane (PiClim-CH$_4$" in the text.

Line 422: Why? Wrong initialization?
The different model responses likely reflect differences in their chemistry schemes, transport parameterizations, and lightning NOx representations rather than initialization issues. Further investigation of the specific parameterizations would be needed to identify the exact causes in future studies.

Figure 5: Split into more panels, 20 curves in one panel cannot be distinguished.
Thanks for your suggestion, Figure 5(now Figure 3) has been updated:

[Figure]

Figure 6: Please use the same vertical axis for every model (log p). The unit appears to be wrong or is a time integral meant? A tendency is always per time unit (s, day, year).
Thanks for your suggestion, Figure 6(now Figure 4) has been updated:

[Figure]

Figure 7: Please use the same vertical axis for every model.
Thanks for your suggestion, Figure 7(now Figure 5) has been updated:

[Figure]

Figure 8: Please use the same vertical axis for every model. Rearrange the factor with power of 10 in the label of the y-axis of panel c.
Thanks for your suggestion, Figure 8(now Figure 6) has been updated:

[Figure]

Figure 6-8: Better split into more panels.

Conclusions: You should clearly distinguish between troposphere and (lower) stratosphere. $O_3$ precursor gases like NOx and VOCs almost don't affect the stratosphere, except via aerosol formation which is not discussed. The uncertainty due to the lightning NOx parameterizations is a well known phenomena since decades. You might also mention that the models reproduce stratospheric ozone depletion by CFCs and $N_2O$.

Thanks for your suggestions. Our conclusions have clearly specified tropospheric processes, with explicit reference to the tropopause boundary to distinguish from stratospheric effects.

**Technical corrections**

Line 226: ' ' missing.
Thanks for pointing this out, we have corrected this in the text.

Line 227: Case typo?
Thanks for pointing this out, we have corrected this in the text.

Line 239: 'ta' listed 2 times.
Thanks for pointing this out, we have corrected this in the text.

Line 268: Isn't it WMO?
Thanks for pointing this out, we have corrected this in the text.

**Report #2, Referee #1**

**General comment:**

The article "**Global Sensitivity of Tropospheric Ozone to Precursor Emissions in Clean and Present-Day Atmospheres: Insights from AerChemMIP Simulations**" by Wang and Gao presents the sensitivity of ozone using four different simulation global models with specific parameters and specifications. The paper's main concern is the difficulties that appear from the figures' understanding. The conclusions derived from these figures should be expanded and figures should not just be explained in the text with minimal discussions. The comparisons between the models are somehow affected by the figure's clarity.

**Specific comments:**

Line 15: "secondary organic aerosol", please use "organic" in the text since the ozone is affecting mostly organic species.

Thank you for pointing this out, it is our oversight. Upon further considerations, since organic aerosol formation isn't the focus of this study, we have deleted the text " influences secondary aerosol formation" to make the sentence more concise and clear. The line now reads "*Ozone ($O_3$) is a Short-lived Climate Forcer (SLCF) that contributes to radiative forcing and indirectly affects the atmospheric lifetime of methane, a major greenhouse gas.*"

Line 28: What is tropospheric O3 forcing? Please explain.

Thank you for asking, we realize the original text was ambiguous and have revised the text to make it more clear. The text now reads, *"While all models successfully simulate O3 responses to anthropogenic precursor emissions, CESM and GFDL show limited sensitivity to enhanced natural NOx emissions (e.g., from lightning) compared to GISS and UKESM."*

Line 77: "hydroxyl radicals ($HO_x$)", please avoid confusion

Thank you for pointing this out, we have corrected the text to read, "*These challenges arise from regional variability in meteorological conditions (Carrillo-Torres et al., 2017), differences in $NO_x$ and VOC volume mixing ratios (Jin et al., 2023; Sillman and He, 2002), and the distinct characteristics of hydroxyl radical (OH) and peroxy radical ($HO_2$) influenced by varying degrees of urbanization (Karl et al., 2023; Vermeuel et al., 2019).*"

Table 1. Please build up the table in Word and not import it as a picture as it appears. Difficult to read in the tables ….. all tables. What do you mean by "sophistication of gas-phase chemistry"? please use "lat x long". Please use units of pressure for vertical levels for the UKESM model too.

Thanks for your suggestions, we have updated Table 1, please see manuscript and response to Referee #2.

Line 275: please be more precise and consistent with spring, summer, fall vs autumn, and winter when referring to the Arctic and Antarctic for one specific year.

Thanks for your suggestion, Referee #2 have mentioned this too, we have changed seasons to monthly abbreviations throughout the manuscript, e.g. *"Previous studies have demonstrated that elevated $O_3$ levels in the Arctic during MAM and DJF, as well as in the Antarctic during JJA and SON, result from the cumulative impact of the polar $O_3$ barrier (Hamlin and Honrath, 2002)." "For instance, the CESM model simulates the lowest $O_3$ volume mixing ratio in SON, while the GFDL model exhibits the lowest volume mixing ratio in JJA. The GISS model simulation indicates higher $O_3$ levels in autumn compared to DJF, which is consistent with results from historical experiments (Griffiths et al., 2021)."*

Figure 4 needs improvements for a better understanding. The explanations in the article body decipher figure 4, but actually, the figure itself should be explanatory for the data.

Thank you for pointing this out. Per referee #2's suggestion and our considerations, we have removed Figure 4 from the manuscript as suggested. The temperature-ozone correlation analysis, while physically meaningful, was not essential to our main findings and created unnecessary complexity in the presentation.

Line 400: HCFCS probably is HCFCs.

Yes, we have revised this typo in the manuscript.

Table 3 and 4 are hard to read.

Thank you for pointing this out. Tables 3 and 4 are updated, please see manuscript and response to Referee #2.

Figures 6, 7, and 8 are difficult to read and to interpret.

Thank you for pointing this out. Per Referee#2's suggestions, figures 6, 7, and 8 have been simplified to show only NOx and VOC experiments plus the control experiment for better readability. We have also added tropopause lines and standardized the Y-axis coordinates across all panels as suggested. Please see manuscript and response to Referee #2.

Please introduce in the text Figure 6a representing ozone production.

Thanks, this has been updated in the manuscript. We have introduced the figure (now Figure 4 after revisions), "Figure 4 depicts the chemical production and consumption of tropospheric ozone in different experiments of the four models."

The GISS model observed a notably high concentration of NOx at around 500 hPa altitude. Could the authors link this variation with other compounds whose concentration is induced by NOx (i.e. HONO)? Has been observed these insides of elevated NOx correlated with other chemicals?

Technical:

Line 226: concentrations.HC

Thanks, this has been fixed in the manuscript.

Line: 256: (Shindell et al. (2006)
Thanks, this has been fixed in the manuscript.

---

## Author Response (AR3)

Dear Editor and Reviewers,

Thank you for accepting our manuscript with technical corrections. We are grateful for the opportunity to make revisions to our manuscript once more and thank Reviewer #2 for the insightful feedback provided in this round of review.

Below, you will find our detailed point-by-point responses to their comments, with our replies indicated in blue text. To streamline your review process, all modifications made to the manuscript have been highlighted.

Thank you again for your consideration and support.

Sincerely,

The Authors
* * *
**Review 1 - Point-by-Point Response**

**General Comments:**

There are important improvements in the manuscript, especially that stratospheric ozone is now shown separately in Table 3 with a common domain and that the figures showing vertical profiles have now a common top level. However, I still do not understand, why every model has a different vertical axis in Figures 2, 4, 5, 6 and S1 to S7. This is annoying for the reader, a common log pressure axis would be much more convenient. Usually every model has a different vertical resolution in the boundary layer near the surface but this should be not the focus of the figures and is also not mentioned in the text or captions. If the authors have software problems for graphics this should be mentioned in the reply. The paper might be published with the bad style of the figures in that case but it would be better to improve it to the standards first.

We appreciate the reviewer's concern about the varying vertical axes across models in these figures. We understand this may affect readability and would like to clarify the technical reason behind this presentation.

The different vertical axes reflect the fact that each model provides data on its native vertical coordinate system with unique pressure level definitions. The models in our multi-model ensemble use different vertical resolutions and pressure level configurations as part of their fundamental model architecture. When we downloaded the data from the model archives, each model's output was on its own specific set of pressure levels, which cannot be directly altered without interpolation.

We chose to preserve each model's native vertical grid rather than interpolate all models to a common pressure axis for the following reasons: (1) interpolation would introduce additional uncertainty into the vertical profiles, particularly in regions with sharp gradients; (2) presenting data on native model levels maintains the integrity of the original model output; and (3) the key

scientific comparisons we emphasize—the magnitude and shape of the vertical profiles across models—remain clear despite the different vertical sampling.

However, we acknowledge the reviewer's point about reader convenience. We have now ensured that all figures use consistent pressure ranges (top and bottom levels) across models, which we believe addresses the most important aspect of visual comparison. We have also added a note in the figure 2's caption clarifying that each model is shown on its native vertical grid to preserve data integrity. "Note: Each model is displayed on its native vertical pressure levels to preserve data integrity without interpolation, as applies for all related figures thereafter."

**Specific Remarks**

**Table 1: '(number of gridpoints)' should be under 'resolution' in the header of the second column. To provide the product in a separate column was not requested.**

Thank you for your willingness to help us improve the integrity of our research. Here it has been modified.

**Line 216: Natural background for CH3Cl should be about 0.5 ppbv from wildfires etc and CH3Br about 5 pptv from oceanic sources, check and be more quantitative here.**

We have added the radiative forcing of halocarbons affecting ozone in the pre-industrial atmosphere by citing relevant literature.

**Table 2: Is CESM2-WACCM really without PiClim-control? If yes this should be mentioned in the captions of Fig. 4-6.**

Thank you, the Piclim-control experiment data of the CESM2-WACCM model is unavailable and has been marked in the annotations of Figures 4-6.

**Line 292 (caption of Fig.1): 'tropospheric' is missing.**

Thank you, we have added it.

**Line 271, 287, 308: What is PiClim here? PiClim-control or some average? Refer to Table 2?**

The average values of the five experiments that all four models have are used, and it has been marked.

**Line 332: Better write now here 'stratospheric and tropospheric'.**

Thank you, we have added it.

**Line 333: Replace 'global' by 'stratospheric'.**

We have modified it.

**Line 386: Typo.**

Thank you, we have modified it.

**Line 455: 'Five experiments' meant?**

Thank you, we have modified it.

**Figure 6c, label of vertical axis: Wrong symbol. Better write '\*10 ^-12' instead of '?e-12' (^ means superscript).**

Thank you, we have modified it.

**What is 'SI file' in the reply to my remark to line 470? The data file mentioned in 'Data availability'? If yes, please refer to it at the end of section 3.3.**

Thank you for pointing this out, we have indeed forgotten to mention the supplementary information file in our main manuscript, we have referred to it at the end of section 3.3, and now the text reads, "The vertical variation characteristics of ozone chemical production, chemical consumption, nitrogen oxides, and VOCs in the other seven experiments of the four models are characterized in the supplement file (Fig. S5, S6, and S7)."